# Computer-Aided Designing Peptide Inhibitors of Human Hematopoietic Prostaglandin D2 Synthase Combined Molecular Docking and Molecular Dynamics Simulation

**DOI:** 10.3390/molecules28155933

**Published:** 2023-08-07

**Authors:** Jing Cui, Yongwei Feng, Ting Yang, Xinglong Wang, Heng Tang

**Affiliations:** 1Wuxi Food Safety Inspection and Test Center, 35-210 South Changjiang Road, Wuxi 214142, Chinaytt20099@163.com (T.Y.); 2Technology Innovation Center of Special Food for State Market Regulation, 35-302 South Changjiang Road, Wuxi 214142, China; 3Science Center for Future Foods, Jiangnan University, 1800 Lihu Road, Wuxi 214122, China; 8202301016@jiangnan.edu.cn; 4Key Laboratory of Bioorganic Synthesis of Zhejiang Province, College of Biotechnology and Bioengineering, Zhejiang University of Technology, Hangzhou 310014, China; 5The National and Local Joint Engineering Research Center for Biomanufacturing of Chiral Chemicals, Zhejiang University of Technology, Hangzhou 310014, China

**Keywords:** inhibitor, hematopoietic prostaglandin D2 synthase, molecular docking, molecular dynamics simulation, Rosetta script

## Abstract

Human hematopoietic prostaglandin D2 synthase (HPGDS) is involved in the production of prostaglandin D2, which participates in various physiological processes, including inflammation, allergic reactions, and sleep regulation. Inhibitors of HPGDS have been investigated as potential anti-inflammatory agents. For the investigation of potent HPGDS inhibitors, we carried out a computational modeling study combining molecular docking and molecular dynamics simulation for selecting and virtual confirming the designed binders. We selected the structure of HPGDS (PDB ID: 2CVD) carrying its native inhibitor compound HQL as our research target. The random 5-mer peptide library was created by building the 3-D structure of random peptides using Rosetta Buildpeptide and performing conformational optimization. Molecular docking was carried out by accommodating the peptides into the location of their native binder and then conducting docking using FlexPepDock. The two peptides RMYYY and VMYMI, which display the lowest binding energy against HPGDS, were selected to perform a comparative study. The interaction of RMYYY and VMYMI against HPGDS was further confirmed using molecular dynamics simulation and aligned with its native binder, HQL. We show the selected binders to have stronger binding energy and more frequent interactions against HPGDS than HQL. In addition, we analyzed the solubility, hydrophobicity, charge, and bioactivity of the generated peptides, and we show that the selected strong binder may be further used as therapeutic drugs.

## 1. Introduction

Prostaglandin D2 (PGD2) is a bioactive lipid mediator that is involved in various physiological processes, including inflammation and immune responses, allergic reactions, sleep regulation, vascular function, and body temperature regulation [1,2,3]. Synthesizing prostaglandins involves a series of reactions initiated by the arachidonic acid (AA) released by membrane phospholipids [4]. AA is subsequently catalyzed by cyclooxygenase into Prostaglandin H2 (PGH2). PGH2 is an intermediate in the synthesis of various prostaglandins, serving as a precursor for the synthesis of various prostanoids, including PGD2, prostaglandin E2 (PGE2), prostaglandin F2α (PGF2α), prostacyclin (PGI2), and thromboxane A2 (TXA2) [5,6]. Hematopoietic prostaglandin D2 synthase (HPGDS) is involved in the synthesis of PGD2 by catalyzing the isomerization of PGH2 to PGD2 [7,8]. Accordingly, HPGDS is mostly expressed in the leptomeninges of the human brain (the membranes covering the brain and spinal cord), which are then secreted into the cerebrospinal fluid as a sleep hormone [9,10,11]. It is also present in immune cells, such as mast cells, macrophages, and T cells, where it contributes to the production of PGD2 during inflammatory responses [2]. HPGDS is highly involved in the inflammatory and allergic pathways for the production of PGD2, which has potent pro-inflammatory effects, including vasodilation and recruitment of immune cells to the site of inflammation. PGD2 is also implicated in allergic reactions, such as asthma and allergic rhinitis, where it contributes to bronchoconstriction and other allergy-related symptoms [12].

Inhibiting HPGDS is a potential strategy for modulating the production of PGD2 and its downstream effects, mainly resulting from inflammation [13,14]. Investigating the inhibitors is a way to cure PGD2-induced inflammation, such as allergic disorders, asthma, sleep disorders, chronic spontaneous urticarial, and hair loss [15,16,17,18]. Studies have been conducted to investigate HPGDS inhibitors, and several commercially available compounds, including HQL-79 [19], TFC-007 [20], and TAS-204 [21], were recognized as HPGS inhibitors. The development of bioinformatics showed great impact on protein engineering and inhibitor design [22,23]. In the past two decades, computer-aided design methods have been implemented in the design of HPGDS inhibitors. These methods are mainly QSAR and molecular docking-based [24,25]. Through QSAR, the key interaction bonds were analyzed and used to evaluate novel compound inhibitors. By using docking-based inhibitor selection, massive compounds were docked to the active pocket to find potential binders with high affinity [26]. Rosetta-based protein design showed high accuracy previously [27,28], while Rosetta FlexPepDock for searching the optimized docking pose of proteins and peptides has shown state-of-the-art performance [29].

Peptide drugs are a class of pharmaceutical agents that consist of short peptides. Peptides can be chemically synthesized or in vivo synthesized. Even though peptides are much shorter than proteins, they still contain key binding regions [30]. Meanwhile, synthetic peptides can mimic the actions of naturally occurring peptides or disrupt specific protein-protein interactions [31]. Peptides can act as agonists, antagonists, or modulators of various biological pathways. Nowadays, peptide drugs have been widely accepted for therapeutic applications. They are used in the treatment of various diseases, including metabolic disorders, cancer, cardiovascular diseases, autoimmune disorders, and infectious diseases [32,33]. Examples of peptide drugs include insulin (used for diabetes), calcitonin (for osteoporosis), and luteinizing hormone-releasing hormone (LHRH) analogs (for hormonal disorders and certain cancers). Previously, inhibitors of HPGDS were identified mainly from chemical compounds; in comparison, this study focused on investigating functional peptides as HPGDS inhibitors as a solution to cure PGD2-related inflammatory responses. We aim to bring out an approach for selecting peptide inhibitors that can outperform chemical compounds.

In this study, we took advantage of Rosetta script and developed a method for generating a peptide library and screening peptides based on molecular docking. This method has been applied to the selection of HPGDS inhibitors. Firstly, peptide library was generated, and every 3-D structure of the peptide was built. Secondly, we localized the binding pocket of HPGDS through a literature review and by observing the interactions between HPGDS and their native inhibitor. Thirdly, we accommodated the designed peptides into the binding pocket according to the position of their native inhibitor. Molecular docking and the evaluation of binding energy were subsequently carried out for the selection of functional peptides. We adopted molecular dynamics simulation for evaluating the inhibitory activity of the designed peptide and its native binder.

## 2. Results and Discussion

### 2.1. Generating Peptide Conformation

Random peptides were initially generated in their sequences and then passed to the Rosetta Buildpeptide module to build their 3-D conformation. In our protocol, the generated peptide was fully refined in its structure and prioritized for docking into the protein receptor (Figure 1A), which mimics the natural protein-peptide interaction order. The full energy-minimized structure was searched using Rosetta CartesianMD. CartesianMD was carried out for 10,000 steps, which account for 20 ps. We show that the optimized structure varied its structure dramatically compared with the native ones. The total score of the peptide before and after structural refinement changed from 1005.637 (average) to 7.656 (average) for all generated peptides, whereas the RMSD difference was 2.76 Å on average (Figure 1B).

It should be noted that the method developed in our study used a fully energy-based method for searching the potential conformation of peptides rather than capturing their structure from protein templates. The reason behind this method was that the peptide can form a different structure when it is apart from a protein structure, and short peptides are less likely to form helices or strands [34].

### 2.2. Molecular Docking

For the docking study, we selected a co-crystalized structure in which HPGDS exhibits inter-molecular binding behavior against a unique inhibitor, HQL [35]. The selected co-crystalized structure (PDB ID: 2CVD) revealed the specific binding pocket on HPGDS, which can guide further molecular docking [35]. Rosetta relax was used for the structural refinement of the HPGDS monomer by removing HQL from the receptor. Our result showed that the Rosetta score shifted from −376.233 to −424.814 in the first round of Rosetta relax (Table 1). The final score after the second round of Rosetta relax was −666.091 (Table 1). The significant difference between the structure with and without refinement confirms that the full-atom minimization step is necessary prior to molecular docking [36]. The native and refined structures showed a distinct RMSD difference of 0.749 Å (Figure 2A).

To perform molecular docking, the generated peptide was accommodated in the binding pocket, where HQL was positioned according to the HPGDS-HQL complex structure. Molecular docking was carried out using Rosetta FlexPepDock [29]. Rosetta InterfaceAnalyzerMover is a tool for evaluating the protein-peptide binding energy, which gives a dG_cross value to represent the binding capacity between protein and peptide. A step-by-step process and validation of our architected Rosetta script were shown in Appendix A. This tool was integrated into our Rosetta script [37], and we extracted the dG_cross value, SASA (solvent accessible surface area, used to represent the buried area in the protein-peptide interface) score, and total score (representing the overall free energy of the protein-peptide complex) for evaluating the interaction between proteins and peptides. The exact dG_cross value, total score, and SASA score are shown in Appendix A.

In this study, we generated 9985 peptides for molecular docking. The generated peptides docked into HPGDS with distinct dG_cross scores ranged from −55 to 5, and the complex total score varied from −179 to −645 (Appendix A). As shown in Figure 2B, the top 10 poses with the lowest dG_cross score (−55 to −52) have a relative low total score (−620 to −608). Few peptides failed to dock into the receptor, including FNPSY, PGWTP, PSAKH, GNYPQ, AEPNM, and QVIIP with high dG_cross scores (Figure 2C). These docking complexes also had high total scores ranging from −346 to −179, indicating hyper-unstable binding complexes (Figure 2C). The SASA analysis of the docked protein–peptide indicates that the SASA score followed the trend of the dG_cross score with a Pearson Correlation Coefficient of 0.54. These results suggest that strong binding is highly corresponded to the larger interface between the protein receptor and the docked peptide.

### 2.3. Comparative Study

The comparative study was used to evaluate the capacity of designed binders by comparing them with their native binder. This method was previously used for validating the designed inhibitor of HPGDS [14]. Due to the fact that the crystalized structure of the HPGDS-HQL complex was not fully refined, calculating the total score for the complex would be meaningless. Therefore, we simply calculated the binding energy between HPGDS and HQL using InterfaceAnalyzerMover (Integrated in Rosetta 2021-16-61629). Our result showed that the dG_cross for the HPGDS-HQL complex was −28.837, which is higher than many of the designed peptides. The two best leads of designed peptides were extracted and analyzed for their interactions with the receptor aligned with the HPGDS-HQL complex. We show that the total number of non-bound interactions between HPGDS-RMYYY and HPGDS-VMYMI was much higher than that of the HPGDS-HQL complex (Figure 3). These results indicate that the best two leads of designed peptides have higher chances of gaining stronger binding with HPGDS than the ligand HQL.

### 2.4. Molecular Dynamics Simulation

A molecular dynamics (MD) simulation was carried out to validate the binding between HPGDS and the designed peptides. This method has previously been implemented in several studies for in silico validation of the designed binders [8,24,26]. In this study, 100 ns of MD simulation was carried out on the two best leads and the HPGDS-HQL complex independently. As shown in Figure 4A, the RMSD values for both HPGDS and the docked ligands and peptides show great differences. It is obvious that HPGDS binds with RMYYY and displays a lower RMSD than that of VMYMI, HQL, and the HPGDS monomer (Figure 4A). The HPGDS-VMYMI complex was unstable during the 40 ns simulation, but its final RMSD value was lower than that of HPGDS-HQL (Figure 4A). The tight binding of protein-ligand can restrict the flexibility of the ligand, which results in less RMSD variations [14,26]. In this study, we showed the complex HPGDS-RMYYY with a lower RMSD than the other two, which indicates that RMYYY had tight binding against HPGDS.

Hydrogen bond formation frequencies were measured to confirm the interactions between proteins and ligands. As shown in Figure 4B, the average hydrogen bonds of HPGDS-RMYYY (5.89 bonds) were much higher than those of HPGDS-VMYMI (1.68 bonds) and HPGDS-HQL (0.29 bonds). These results confirm the docking result presented in Figure 3. The interactions between HPGDS and the peptides RMYYY and VMYMI were mainly hydrogen bonds, but for HPGDS-HQL, the bond type was mainly Pi-Pi stacking or Pi-alkyl interaction. Further, the key binding area of the protein was analyzed, as shown in Figure 4C. Binding to peptides of RMYYY and VMYMI was supposed to stabilize the binding area against ligands and peptides, as the RMSF values of RMYYY for residues 35–50 and 104–113 were less than the other two complexes. Additionally, the protein monomer without any binding partner displays a much higher RMSF value for residues 35–50 and 104–113, indicating the binding pocket has high flexibility for attracting substrates while in the non-binding mode (Figure 4C). These results support the idea that residues 35–50 and 104–113 are key for protein-ligand binding.

### 2.5. Identification of Key Binding Residues

The identification of key residues for protein-ligand binding was carried out using RMSD-based cluster analysis and MMPBSA. Cluster analysis is used to process a large group of structures and sort them into smaller groups. The RMSD-based cluster analysis implemented in this study exported clusters for HPGDS-HQL, HPGDS-RMYYY, and HPGDS-VMYMI independently. We used the top two clusters of every trajectory to represent the binding behavior between proteins and ligands. Notably, the top 1 cluster occupied 15.42%, 25.55%, and 38.99% for HPGDS-HQL, HPGDS-RMYYY, and HPGDS-VMYMI using a RMSD cutoff of 0.16 nm, suggesting these clusters are representative for the independent simulations. As shown in Figure 5, hydrogen bonds were the dominant non-bound interactions for the complexes of HPGDS-RMYYY and HPGDS-VMYMI, whereas HPGDS-HQL was a Pi-Pi stacking or Pi-alkyl interaction. Cluster analysis was also conducted on the receptor monomer, which shows that the binding pocket was occupied by the peptide in the inbound state (Figure 5B).

To confirm the key residues for protein-ligand binding, MMPBSA was carried out using the last 10 ns of every trajectory. Independent analysis revealed that several residues were important for receptor and ligand binding with much lower binding energy, such as R14 and F15 for the HPGDS-RMYYY complex and D96, T159, Y152, and M99 for the HPGDS-VMYMI complex. Combinatory analysis indicates that residues such as M99 and R14 were present in two of the binding complexes with relative low binding energy (Figure 6).

### 2.6. Analyzing Peptide Properties

The physical and chemical properties are important for evaluating peptides as potential therapeutic drugs. In this study, we analyzed the solubility [38], charge, hydrophobicity [39], and bioactivity (PeptideRanker) [40] of the generated peptides. Our result indicates that strong binding peptides were more likely to be insoluble and highly hydrophobic, and their solubility and hydrophobicity are weakly correlated with binding score (Figure 7). But, the binding score shows non-correlation with peptide charges, indicating the hydrophobicity of the generated peptides was critical for protein-peptide binding, rather than peptide charge. Additionally, strong-binding peptides tend to have high bioactivity (Figure 7). Most of the best 10 leads had high hydrophobicity and low solubility, but their bioactivity was relative higher, which displayed a score of more than 1 (Figure 7). The two best-selected lead peptides, RMYYY and VMYMI, were soluble and insoluble, respectively, and their bioactivity was similar (Figure 7), suggesting RMYYY may be a better drug candidate.

## 3. Materials and Methods

### 3.1. Docking Receptor Preparation

The X-ray crystal structure of HPGDS carrying its native inhibitor compound HQL (PDB ID: 2CVD) was selected [35]. The given structure displayed a high resolution of 1.45 Å. The structure of HPGDS was prepared by removing the water molecule and the other compounds, including HQL. We adopted two rounds of Rosetta relax for refining the structure of HPGDS [41]. The two rounds of Rosetta relax aim for repacking the protein sidechains and conducting all-atom minimization, respectively (Appendix A). The Rosetta score function ref2015 was used to represent the folding energy of the minimized protein [42], and the achieved total score from ref2015 was used to guide the selection of the proper structure. During each round of Rosetta relax, we adopted the minimized structure with the lowest total score for the next step.

### 3.2. Building a Random Peptide Library

For the selection of peptide binders, we organized a peptide library by firstly generating random peptide sequences and, secondly, building its 3-D structure using Rosetta BuildPeptide [37]. A peptide can form a special conformation in its solubilized state. Meanwhile, the secondary structure of peptides can be changed in their inbound and unbound states [34]. Therefore, to capture the proper 3-D conformation of the generated peptides, we adopted Rosetta Monte Carlo simulation for refining the peptide conformation. CartesianMD was carried out for 10,000 steps, which completely account for 20 ps (Appendix A), and the score function talaris2014_cart was used for evaluating the total score of the output minimized peptides. In this study, the total number of non-redundant peptides generated was 9985 (Appendix A).

### 3.3. Integrating Rosetta Script for Protein-Peptide Docking

We adopted the Rosetta script for integrating the molecular docking protocol. In our protocol, the position of the native binder compound HQL was considered to accommodate novel peptides. Through inspecting the co-crystallized structure of HPGDS with HQL, the coordinates of O1 in HQL were selected for accommodating the generated peptides (Figure 8). Dragging the peptide was performed using Gromacs-2020 [43]. The inbound structure was used as input for the Rosetta script to perform molecular docking. For the docking protocol, optimized peptides were refined for their interaction with protein receptors using FlexPepDock [29]. Then, the backbone and sidechains were minimized using Minmover, followed by all-atom refinement using FastRelax. Finally, we adopted InterfaceAnalyzerMover for evaluating the interface binding energy of the docking pose (Appendix A).

### 3.4. Molecular Dynamics Simulation

MD simulation is a common way of analyzing the binding behavior and investigating the binding mechanism of protein-peptide complexes. To validate the two best leads, MD simulation was independently carried out on HPGDS-RMYYY, HPGDS-VMYMI, and HPGDS-HQL. We used the docking complex obtained from Rosetta dock and subjected it to MD simulation using Gromacs-2020 [43]. The simulation system was immersed with the SPC/E solvent model in an orthorhombic box. The system was neutralized using Na^+^ and Cl^−^. The protein was placed at a distance of 15 Å from the edge of the simulation box. The simulation system was minimized using the steepest descent method for initial energy minimization and equilibrated using the isochoric–isothermal ensemble and isothermal–isovolumetric ensemble under 300 K for 100 ps, respectively. We used a time step of 2 fs during the simulation. The MD simulation was conducted for 100 ns, and the simulation trajectories were collected for analysis.

### 3.5. MMPBSA Analysis

The Molecular Mechanics Poisson-Boltzmann Surface Area (MMPBSA) is a robust method for evaluating the free energy between molecules. This method was implemented to investigate the critical sites of protein-ligand binding. We adopted gmxMMPBSA for MMPBSA analysis by extracting the trajectories of the last 10 ns of the simulation [44].

### 3.6. Peptide Properties Calculation

The generated peptides were calculated for their properties, including solubility, charge, and hydrophobicity, using the Python peptides module. In addition, we used PeptideRanker (http://distilldeep.ucd.ie/PeptideRanker/, accessed on 29 July 2023) to predict the bioactivity of the generated peptides.

## 4. Conclusions

This work introduced a method for designing peptide binders, which was applied to HPGS for selecting its potential binders. By using our approach, a peptide library containing 9985 random 5-mer peptides and their conformations was used for molecular docking. We integrated the Rosetta script to perform molecular docking and calculate the binding energy. The two best leads of designed peptide binders were validated and compared with the native compound binder HQL. We show that the designed peptides have higher chances of interacting with the receptor HPGS and have stronger binding energy compared with HQL. Through peptide property analysis, our results show that the peptide candidates, including RMYYY, MARYI, DYQFI, and ERMNM, with strong binding against HPGS also display high solubility. In addition to computational study, these candidates still need further experimental validation to prove their capacity as HPGS binders.

## Figures and Tables

**Figure 1 molecules-28-05933-f001:**
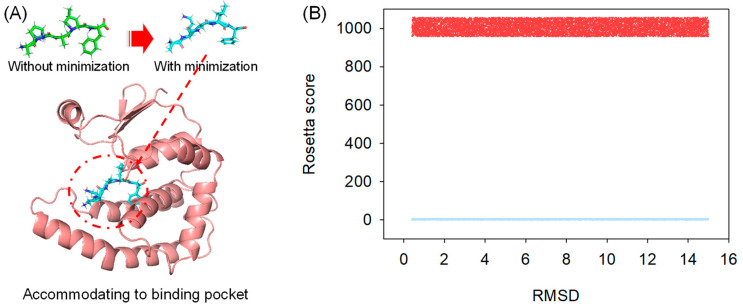
Peptide structural refinement. (**A**) Structural refinement applied to peptides to prioritize molecular docking; (**B**) The calculated Rosetta total score of peptides before (Red) and after (Blue) structural refinement; the RMSD difference of every peptide was shown.

**Figure 2 molecules-28-05933-f002:**
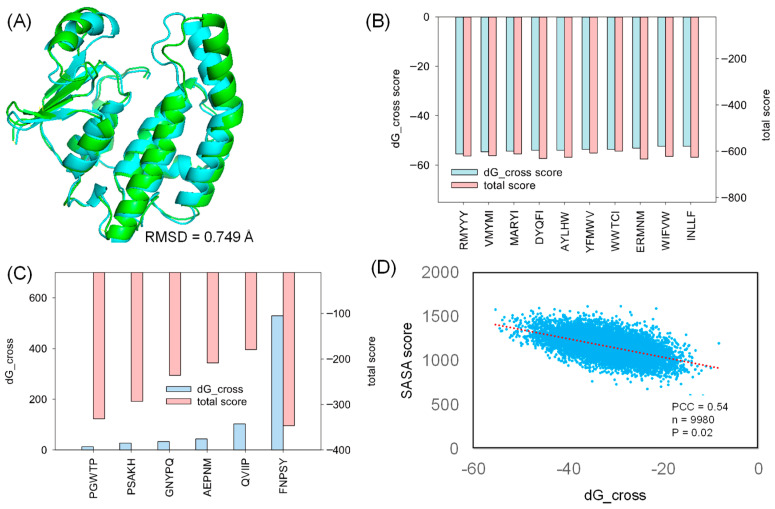
The strong and weak binding of designed peptides against HPGDS. (**A**) The RMSD difference between HPGDS with (green) and without (cyan) structural refinement. The dG_cross and total score were calculated using Rosetta for the strongest (**B**) and weakest (**C**) docking poses. (**D**) The obtained SASA score after sorting the dG_cross score from small to large. PCC: Pearson Correlation Coefficient; n: sample size; P: *p*-value.

**Figure 3 molecules-28-05933-f003:**
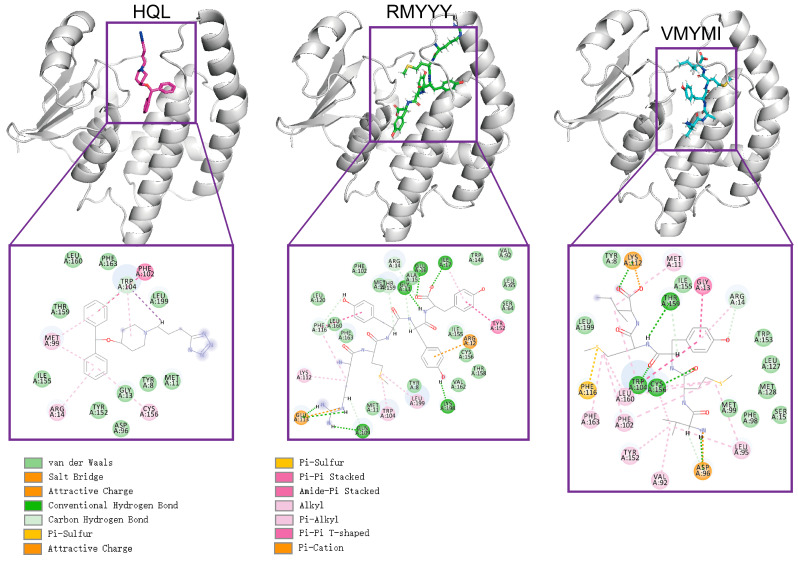
Visualization of protein-ligand interactions after molecular docking. Molecular docking was carried out using Rosetta script integrated with FlexPepDock; the 3-D structure shows the relative position of protein and ligand, whereas the 2-D structure shows the exact non-bound interaction between protein and ligand.

**Figure 4 molecules-28-05933-f004:**
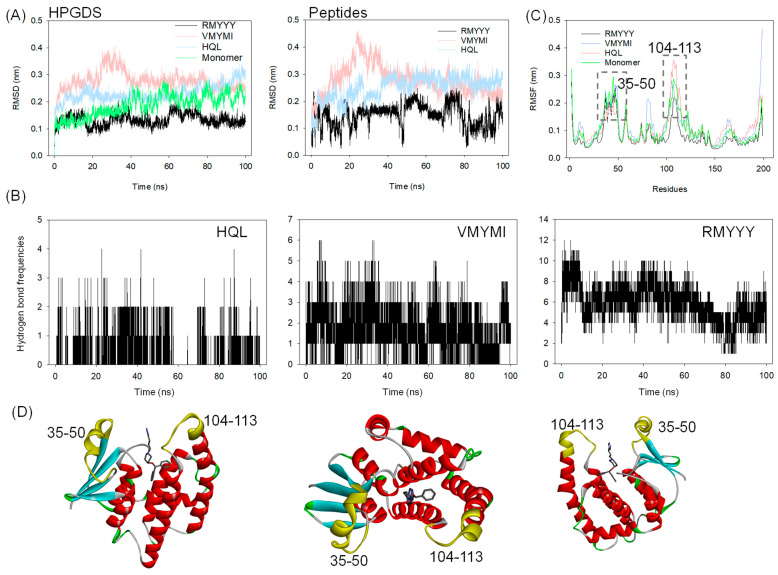
Molecular dynamics simulation of HPGDS binding with selected ligands. RMSD (**A**) and RMSF (**C**) analysis of HPGDS in the docking complex with compound HQL and peptides RMYYY and VMYMI; (**B**) Hydrogen bond formation frequencies between protein and ligand during MD simulation. RMSD-based cluster analysis of MD simulation trajectory, the portion of cluster was labeled; (**D**) Structural representation of the position of high-flexible residues and ligands.

**Figure 5 molecules-28-05933-f005:**
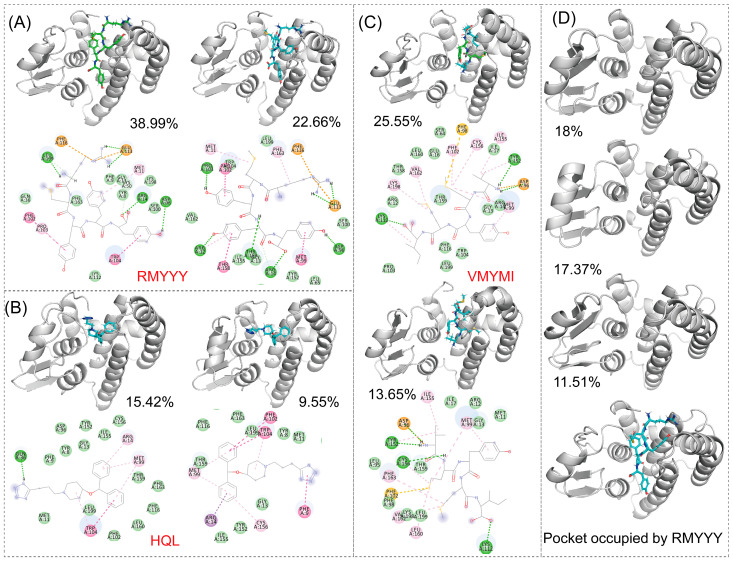
RMSD-based cluster analysis. RMSD-based cluster analysis conducted using the trajectory of HPGDS carried three ligands and the HPGDS monomer independently. Protein binding with different peptides or ligands (**A**–**C**). The protein monomer is shown in (**D**). The portion of the cluster is shown in figure.

**Figure 6 molecules-28-05933-f006:**
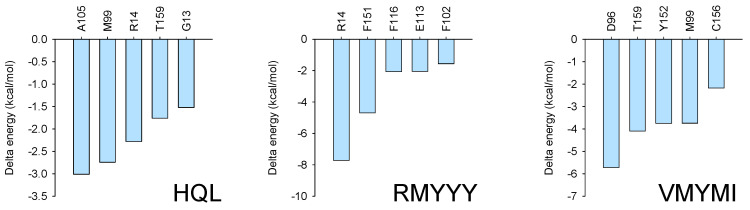
MMPBSA analysis. MMPBSA analysis was conducted using gmxMMPBSA, which extracted the last 10 ns of every trajectory for the binding energy calculation.

**Figure 7 molecules-28-05933-f007:**
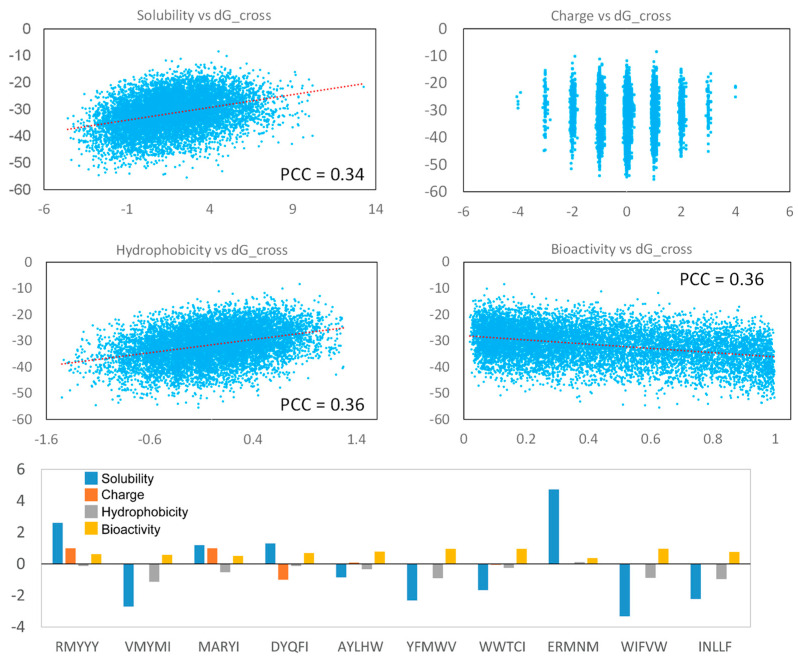
Peptide property analysis. The Pearson Correlation Coefficient of each figure was calculated using the Excel linear regression module. The parameters for the top 10 peptides are shown in the Histogram.

**Figure 8 molecules-28-05933-f008:**
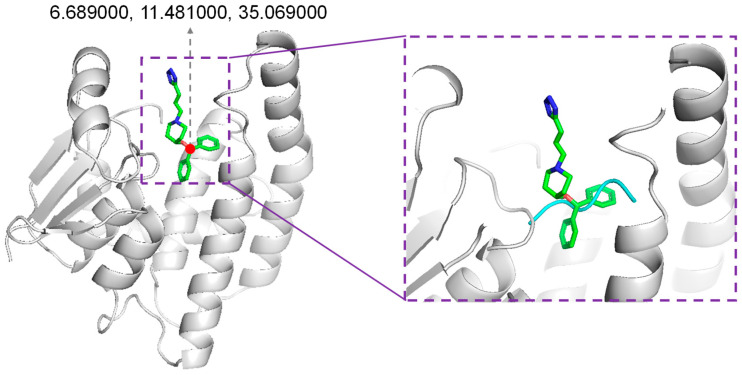
Accommodating generated peptides. The position for accommodating generated peptides was the original position of compound HQL in the co-crystalized structure (PDB ID: 2CVD).

**Table 1 molecules-28-05933-t001:** The Rosetta score obtained from two rounds of Rosetta relax on CypD.

	1st Round	2nd Round
nstruct_1	−423.413	−657.422
nstruct_2	−423.324	−660.113
nstruct_3	−423.27	−666.091
nstruct_4	−423.159	−657.826
nstruct_5	−423.096	−665.231
nstruct_6	−423.486	−664.487
nstruct_7	−424.494	−662.3
nstruct_8	−424.38	−664.144
nstruct_9	−424.675	−662.181
nstruct_10	−424.429	−659.845
nstruct_11	−424.675	−662.543
nstruct_12	−424.704	−659.206
nstruct_13	−423.319	−662.416
nstruct_14	−424.675	−661.017
nstruct_15	−424.543	−665.381
nstruct_16	−424.733	−661.849
nstruct_17	−424.197	−661.809
nstruct_18	−424.163	−664.934
nstruct_19	−424.733	−661.004
nstruct_20	−424.675	−664.397
nstruct_21	−423.413	−660.583
nstruct_22	−424.675	−668.228
nstruct_23	−424.143	−660.682
nstruct_24	−424.145	−662.245
nstruct_25	−424.068	−665.338
nstruct_26	−423.234	−664.834
nstruct_27	−424.149	−661.667
nstruct_28	−424.335	−663.85
nstruct_29	−424.775	−658.93
nstruct_30	−424.675	−658.555
nstruct_31	−424.647	−664.356
nstruct_32	−424.675	−665.182
nstruct_33	−424.068	−664.331
nstruct_34	−424.773	−664.757
nstruct_35	−424.675	−663.109
nstruct_36	−424.814	−662.415
nstruct_37	−424.733	−663.211
nstruct_38	−424.733	−665.031
nstruct_39	−424.659	−665.633
nstruct_40	−423.159	−665.943
nstruct_41	−423.128	−663.531
nstruct_42	−423.413	−662.742
nstruct_43	−424.675	−665.409
nstruct_44	−423.138	−664.335
nstruct_45	−423.267	−658.936
nstruct_46	−423.267	−660.002
nstruct_47	−425.471	−658.008
nstruct_48	−424.675	−664.131
nstruct_49	−424.742	−658.943
nstruct_50	−424.741	−665.313

## Data Availability

The authors declare that all data supporting the findings of this study are available in the article and its Appendix A or are available from the corresponding author on request. The code for conducting this study is provided in Appendix A. The generated peptides in this study and their properties are provided in Appendix A.

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
