# Peer review of "Computer-Aided Designing Peptide Inhibitors of Human Hematopoietic Prostaglandin D2 Synthase Combined Molecular Docking and Molecular Dynamics Simulation"

_molecules, 2023, doi:10.3390/molecules28155933_

Round 1
Reviewer 1 Report
In this paper, Cui et al. carried out molecular docking and molecular dynamics simulation for designing peptide inhibitors of human hematopoietic prostaglandin D2 synthase. Here are some comments on this paper:
1. In the Supplemented file S1 ##Rosetta script for molecular docking, to my understanding, the <RESIDUE_SELECTORS> should not be empty. At least, the docking sites and the layers of the receptor should be clarified, and then be passed to the <MOVERS>. If the residue selectors are empty, how can the peptide can dock to the original active binding sites?
2. According to the S2, the energy ranking of peptides, there were only minor differences in the Dg_cross scores of the top 4 peptides. What’s more, the total score is in a big difference. The authors only choose the 2 peptides don’t make much sense to me. Additionally, I suggest taking more matrices in to filter the potential peptides, such as SASA, binding shape complementary, etc.
3. The physical and chemical properties are also very important and are not considered at all, for example, the prediction of the solubility of the peptide.
4. To the MD studies, the author studies the RMSD and RMSF of the complex. I believe the dynamic conformation of the independent receptor is also very important, which should be demonstrated. Whether the active structural integrity is maintained when the peptide is bound.
Author Response
Thank you for your positive comment. The manuscript was revised as suggested. The answer to each comment is listed in the revised file, and the revised parts in the updated manuscript are highlighted with yellow.

Reviewer 2 Report
Review Report:
Manuscript Number: molecules-2536666
Title: Computer aided designing peptide inhibitors of human hematopoietic prostaglandin D2 synthase combined molecular docking and molecular dynamics simulation.
Recommendation: Minor revision
I carefully reviewed the paper entitled “Computer aided designing peptide inhibitors of human hematopoietic prostaglandin D2 synthase combined molecular docking and molecular dynamics simulation" submitted by Jing Cui et al. As a referee, I have carefully reviewed the manuscript, and I must commend the authors for their well-written article and the clear motivation behind their research. This study is of great importance and, with some minor revisions, could significantly contribute to the scientific literature and benefit fellow scientists. Below are the points that need to be addressed:
1. The authors would mention shortly the mode of synthesis in the abstract.
2. It is necessary to increase and update the number of references by adding some latest references. What is the role of Human Hematopoietic Prostaglandin D2 Synthase (HPGDS) in physiological processes, and why is it important to investigate inhibitors of HPGDS?
3. The combination of molecular docking and molecular dynamics simulation is a widely used approach to study protein-ligand interactions and predict binding affinities. What is the methodology used for selecting the protein in molecular docking studies.
4. Why the authors choose the particular protein for docking analysis? Which amino acids interacts more with the drug. Are they strong interactions? Is there any PASS analysis performed? Give the correct reference to the original of the pdb structure used.
5. What were the observed biological activities of the synthesized derivatives? Did they exhibit any noteworthy properties or potential applications? Explain in detail about the table data of docking.
6. What implications do the results of this study have for the development of potential anti-inflammatory agents targeting HPGDS? Were there any limitations or challenges encountered during the molecular docking and simulation process, and how were they addressed?
7. To correct grammatical errors, it should be read by someone with good English and grammatical errors should be corrected.
By minor revision, I am of the opinion that the article can suitable for publication in "Molecules" journal.
The manuscript should be polished.
Author Response

(The authors gave the same response as above.)

Round 2
Reviewer 1 Report
Thank you very much for the authors’ response and modifications to the paper! After carefully reviewing the authors’ response and the revised manuscript. I found that the authors have responded to my concern and I suggest this manuscript could be accepted.